# Community-directed treatment with ivermectin in Maridi, South Sudan: Impact of an onchocerciasis awareness campaign and bi-annual treatment on therapeutic coverage

Joseph Nelson Siewe Fodjo[1], Moses Okwii[2], Stephen Raimon Jada[3], Amber Hadermann[1], Jacopo Rovarini[3], Luis-Jorge Amaral[1], Rogers Nditanchou[1,4], Yak Yak Bol[5], Makoy Y. Logora[5], Jane Y. Carter[6], Johan Willems[7], Robert Colebunders[1,8]*

1 Global Health Institute, University of Antwerp, Antwerp, Belgium, 2 Kampala University, Juba, Republic of South Sudan, 3 Amref Health Africa, Juba, Republic of South Sudan, 4 Sightsavers, Yaoundé, Cameroon, 5 Neglected Tropical Diseases Unit, Ministry of Health, Juba, Republic of South Sudan, 6 Amref Health Africa Headquarters, Nairobi, Kenya, 7 Christoffel-Blindenmission/ Christian Blind Mission e.V. (CBM), Bensheim, Germany, 8 Department of Tropical Disease Biology, Liverpool School of Tropical Medicine, Liverpool, United Kingdom

* robert.colebunders@uantwerpen.be

## Abstract

### Background

A high onchocerciasis disease burden and a low coverage of community-directed treatment with ivermectin (CDTI) have been observed in many parts of South Sudan. In the Maridi County, CDTI was re-introduced in 2017 and various interventions implemented to improve coverage.

### Methods

Through successive community-based surveys, we investigated whether an onchocerciasis awareness campaign and a switch from annual to bi-annual distribution of ivermectin in Maridi County increased CDTI coverage. We also reviewed the evolution of ivermectin distribution in Maridi since 2017 and identified the determinants for ivermectin uptake.

### Results

For past years in Maridi, CDTI programme performance has been highly variable due to security concerns, limited funding, misconceptions about ivermectin, and poor organisation of mass treatment campaigns. Community-based surveys conducted between 2018 and 2024 in Maridi found that upon switching from annual CDTI (2017–2019) to bi-annual CDTI (2021 onward), therapeutic coverage significantly increased from 40.8% in 2017 to 70.3% in 2023. Lower age, male gender, more CDTI

**Data availability statement:** DATA WERE UPLOADED ON ZENODO doi 10.5281/zenodo.16983738.

**Funding:** This work was supported by a Research for Health in Humanitarian Crisis (R2HC) grant awarded to Amref Health Africa (project 78719). JNSF received funding from the Research Foundation–Flanders (FWO) (grant 1296723N). RC is supported by the FWO (grant G0A0522N). L-JA had funding from the La Caixa Foundation, grant number B005782. The funders had no role in the study design, data collection and analysis, decision to publish, or manuscript preparation.

**Competing interests:** The authors have declared that no competing interests exists.

information sources, and awareness of a link between onchocerciasis and epilepsy were all associated with increased uptake of ivermectin.

## Conclusion

This study showed that with reinforced awareness raising accompanying biannual CDTI, a higher ivermectin treatment coverage is achievable. The findings present an opportunity for the health system to advance its onchocerciasis elimination scheme in remote, conflict-stricken communities in South Sudan.

## Author summary

The neglected tropical disease onchocerciasis (also known as river blindness) is endemic is several African countries including South Sudan. For decades, the World Health Organization has relied on Community-directed treatment with ivermectin (CDTI) as the mainstay the public health intervention against onchocerciasis in endemic communities. For CDTI to be effective, at least 80% of the community must receive the drug annually for 10–15 consecutive years. Deploying and sustaining ivermectin distribution at an acceptable coverage has been challenging in the Maridi County of South Sudan, which has been subjected to insecurity and humanitarian crises for several years. Our study documents how upon re-introducing CDTI in 2017 after a period of interruption, coverage was initially low at 40%. Following a series of activities including community awareness campaigns and a switch from annual to bi-annual CDTI in Maridi, coverage rose to 70% in 2023 implying that more people were taking the drug each year. Awareness of a link between onchocerciasis and epilepsy was one of the motivating factors for ivermectin uptake in the study villages. Implementing similar approaches in other humanitarian and conflict-stricken settings could help accelerate onchocerciasis elimination from endemic communities.

## Background

Onchocerciasis, also known as river blindness, is caused by the filarial worm *Onchocerca volvulus*. The disease is present in 31 African countries, the Arabian Peninsula (Yemen) and the Americas (two remaining foci in the Amazon area) [1] and is the worlds second leading cause of preventable blindness [2]. It is estimated that 99% of the 20.9 million *O. volvulus*-infected individuals live in Africa. Of these, 14.6 million are considered to have onchocerciasis induced skin disease and 1.15 million to have vision loss [3]. Increasing epidemiological evidence also suggests that onchocerciasis may induce epilepsy [4].

 Onchocerciasis elimination programmes mainly rely on community-directed treatment with ivermectin (CDTI) [3,5]. Ivermectin is a microfilaricidal drug that kills *O. volvulus* embryonic larvae (microfilariae) and temporarily sterilises adult worms

[6]. Using the CDTI approach, the African Programme for Onchocerciasis Control (APOC) has been successful in controlling onchocerciasis in several African countries [7]. However, many onchocerciasis-endemic areas in Africa – particularly those that have experienced periods of insecurity and where CDTI coverage has been sub-optimal or interrupted – are still characterized by high onchocerciasis infection and transmission, and in some places a high prevalence of onchocerciasis-associated morbidity including onchocerciasis-associated epilepsy (OAE) [4,8–10] as is the case in South Sudan [11–13].

South Sudan is among the countries in Africa with a high burden of onchocerciasis, with about half (49%) of the country reported to be endemic for the disease [14] despite many years of ivermectin Mass Drug Administration (MDA). As early as 1995, non-governmental organisations (NGOs) such as HealthNet International (HNI) were implementing treatment with ivermectin [15]. The National Onchocerciasis Control Programme of South Sudan, established in 1996, initiated CDTI activities by 2004 [16]. In 2015, the national Ministry of Health (MOH) established the Onchocerciasis Control and Elimination Programme with the objective of interrupting onchocerciasis transmission in 80% of endemic areas in the country by 2020 [16]. At present, this target is far from being achieved mainly due to frequent security concerns hampering the yearly implementation of CDTI, limited funding, as well as inadequate numbers and insufficient supportive supervision of community drug distributors (CDDs) [17,18]. In Maridi County, Western Equatoria, State, CDTI was introduced in the early 2000s [19]. However, both therapeutic coverage (the proportion of people taking the drug within a community) and geographic coverage (the proportion of endemic communities reached by the programme) remained suboptimal for onchocerciasis elimination. In the programme's early years, therapeutic coverage often fell below the WHO-recommended threshold of 65% required to eliminate onchocerciasis as a public health problem [18,19]. In 2006, around 40% (4.1 million) of the South Sudan's population were at risk of onchocerciasis, of which 3.6 million were eligible for CDTI [15]. However, only 26% of the eligible population received treatment with ivermectin in the five CDTI areas [15,20]. In 2016, CDTI activities were interrupted due to the transition of programme management from APOC to the MOH [21]. This period coincided with a shift in the WHO's objective from eliminating onchocerciasis as a public health problem to achieving transmission elimination; the country therefore adopted a minimal therapeutic coverage goal of 80% of the entire population [16]. CDTI was reintroduced in 2017.

There have been recent reports of high Ov16 seropositivity rates among children under 10 years of age in Maridi, Mundri West, and Mvolo Counties in Western Equatoria State of South Sudan [12,13,22]. In Maridi in 2019, Ov16 seroprevalence among children living at a site close to a dam with nearby blackfly vector breeding sites was 40.0% for those aged 3−6 years and 66.7% among 7−9-year-olds [22]. In addition, the overall epilepsy prevalence in eight study sites in Maridi was 4.4%, with an epilepsy prevalence of 11.9% at the site closest to the Maridi dam which harbours the blackfly breeding site [11]. Based on these findings and as advocated by some NGOs working in the area (notably the Christian Blind Mission (CBM) and Amref Health Africa (Amref)), the CDTI regimen was switched from annual to biannual (every six months) in Maridi. However, in 2020, CDTI was temporarily interrupted throughout South Sudan owing to the COVID-19 pandemic but resumed in February 2021. Maridi additionally benefited from a second ivermectin distribution round in August 2021.

As a prelude to biannual CDTI resumption in 2021, Amref supported the government in an exercise to confirm existing CDDs or appoint new ones in endemic communities, ensuring the selected individuals were well respected residents of their respective villages and had received the endorsement of the village chiefs. To reinforce CDTI participation, in January 2021 the MOH launched a national onchocerciasis and lymphatic filariasis (LF) awareness and mobilisation campaign. Activities conducted during this campaign included:

1) *Village and road announcements*: Community mobilisers were assigned to inform residents about onchocerciasis and LF in their villages. The basic messages included information about onchocerciasis, LF morbidities and disabilities, the importance of taking ivermectin to prevent disease, and the safety of ivermectin. Announcements were made at markets and meeting places using megaphones, and in the streets using hired vehicles with loudspeakers.

2) *Radio talk-shows*: Radio talk-shows were held with the participation of the county's Neglected Tropical Diseases Focal Person and the national supervisor of the campaign.

3) *Use of posters*: Posters were conspicuously displayed in public places. These posters were mostly visual and displayed pictures of the signs and disabilities caused by onchocerciasis and LF, and the number of ivermectin tablets to be taken during CDTI campaigns.

The onchocerciasis awareness campaign was intensified in Maridi County by the Amref research team. Meetings were organised to sensitise and mobilise health professionals, the local government, head teachers, religious leaders from churches and mosques, youth association representatives, and local NGOs. Moreover, community meetings were held to reach wider audiences in villages, particularly those with limited access to radio. Awareness messages to encourage people to take ivermectin were also delivered during football matches and during church services. The campaign activities in Maridi were monitored daily by the Amref research team and the County Chief Mobiliser, who addressed the challenges faced by CDDs. The County Chief Mobiliser visited the villages frequently to provide direct supervision and maintain communication with the mobilisation team in villages as well as using the telephone network. In this paper, we assess the therapeutic coverage of ivermectin treatment in Maridi before and after conducting the awareness campaign and implementing biannual CDTI.

## Methods

### Study setting

Field studies were conducted in Maridi County located in Western Equatoria State, South Sudan (Fig 1). As of 2017, the entire Maridi County had an estimated population of about 106,834 people [23]. The population in Maridi County is ethnically diverse, including groups such as the Baka, Mundu, Avukaya, Zande, Moro Kodo and Wetu. The area is known for its mango trees which thrive during the rainy season. The state features abundant rivers with fast-flowing water, such as the Maridi River, Yei River and Naam River, creating ideal breeding environments for blackflies, the vectors responsible for transmitting onchocerciasis. Farming, fishing, and animal husbandry (cattle, goats, sheep) are the main livelihood activities in the region. Maridi town is the County's administrative centre. Since 2018, a joint team consisting of Amref staff,

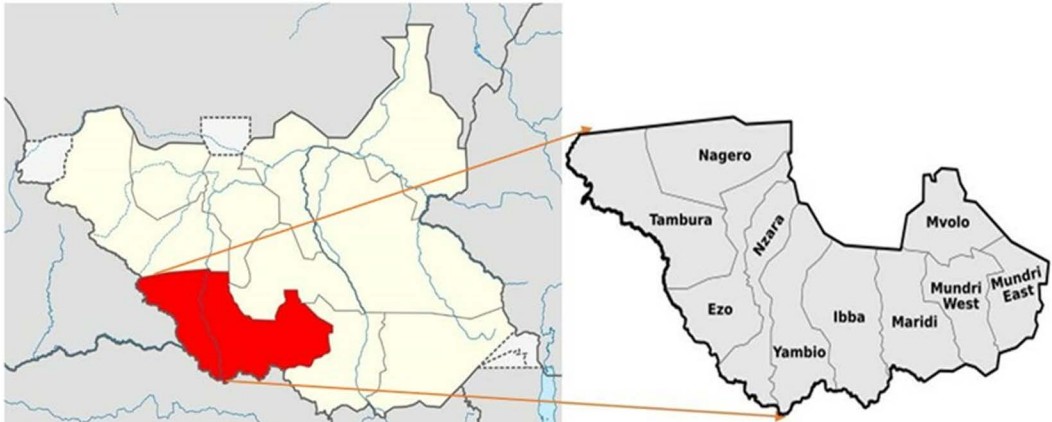

**Fig 1. Location of Maridi County within South Sudan (left) and Western Equatoria State (right) [24].** This file is made available under the Creative Commons CC0 1.0 Universal Public Domain Dedication. Available on Wikimedia page: https://commons.wikimedia.org/wiki/File:Western_Equatoria_State_Counties.svg.

together with the MOH and researchers of the University of Antwerp (Belgium) has been investigating the link between onchocerciasis and epilepsy in Western Equatoria State with a focus on the Maridi area.

## Study procedures

Different ivermectin coverage evaluation surveys were performed at different time intervals in Maridi County. Fig 2 provides a visual timeline of the evolution of CDTI in Maridi, considering the various interventions to boost CDTI adherence and the coverage assessments in the study sites. In this study, ivermectin coverage refers to the therapeutic coverage achieved by CDTI, i.e., the proportion of the entire village population that took the drug during a given distribution session. Our research team found an ivermectin coverage of 40.8% in Maridi in 2017 (considered as the baseline for this study) [11].

1.  *Community-based surveys in Maridi County, 2018–2024*

In 2018, 2022, and 2024, the Amref research team conducted cross-sectional door-to-door surveys in the Maridi Central area to document CDTI coverage. The study site (including the localisation of the Maridi dam, main blackfly breeding site in this area) and the methodology of these house-to-house surveys has been described in detail in previous publications describing the epidemiology of onchocerciasis-associated epilepsy in the area [11,25]. During these surveys, all household members were asked about ivermectin intake during the most recent CDTI round(s) (S1 File). For family members who were unable to answer (e.g., too young, disabled) or absent during the survey, the information was provided by another household member.

For convenience, the 2024 ivermectin coverage survey was limited to only three study sites in the Maridi County namely: Hai-Gabat, Kazana-1, and Kazana-2. These sites were chosen to be representative of high transmission communities

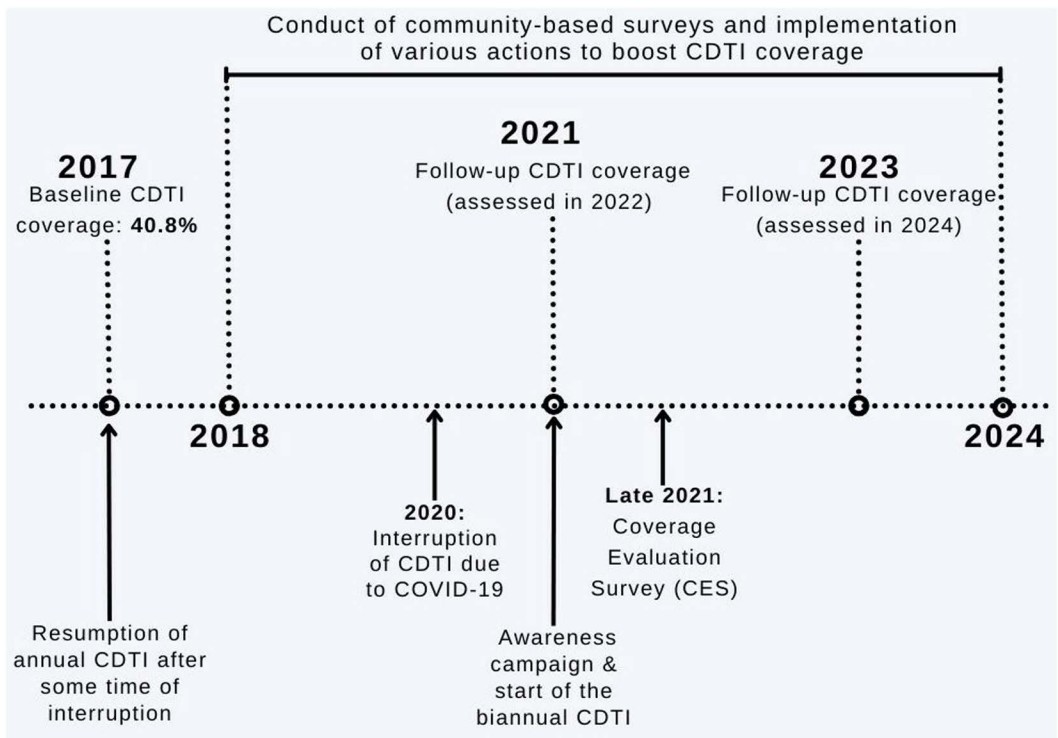

**Fig 2. Assessment of the performance of CDTI in Maridi between 2017–2023.**

(Kazana-1 and Kazana-2) and also a lower transmission community (Hai-Gabat). During this latest survey, participants who responded "no" to ivermectin intake were asked to state the reasons why they did not take the drug. We also sought to identify individuals who systematically refuse to take ivermectin in the study communities, and the reasons for this.

2. *Independent Ivermectin Treatment Coverage Evaluation Survey (CES) in Maridi, 2021*

This survey was conducted by an independent research team from Kampala University in November 2021, less than three months after the CDTI round in August 2021. Probability Proportionate to Estimated Size (PPES) sampling methodology recommended by the World Health Organization (WHO) [26] was used to select participants. In brief, 50 households within each of the 30 sub-units or villages in the Maridi CDTI evaluation unit were selected for participation based on a predetermined sampling interval. These households were visited by trained data collectors who administered questionnaires using electronic device. Sociodemographic information and ivermectin intake history were collected for all household members who were residing in the village at the time of the survey.

3. *Ivermectin coverage survey in 2023 among children and their parents or guardians in Maridi*

In February 2023, a pilot ivermectin coverage survey was conducted by Amref only among children aged 3−9 years and their parents or guardians in Maridi. This survey was nested within a larger study which assessed onchocerciasis transmission following the implementation of various elimination measures. The children and their parents or guardians met with the research team at an agreed location in each village and were interviewed regarding their ivermectin intake during the previous CDTI round (August 2022).

## Statistical analysis

The collected data were noted on paper forms and entered into electronic spreadsheets, cleaned, and analysed in R version 4.4.1. CDTI coverage was calculated as the percentage of surveyed participants who reported taking ivermectin. Additionally, age-specific ivermectin coverage was assessed among participants aged ≤20 years due to their increased vulnerability of developing OAE [4]. Comparisons of CDTI coverage across study years was done using the Pearson's chi-square test and considered the proportions of surveyed individuals who took ivermectin at least once during a given year, regardless of whether the CDTI regime was annual or biannual. Multivariable logistic regression was used to investigate preselected plausible factors that could be associated with ivermectin intake in August 2021, after various interventions had been implemented to boost CDTI coverage in Maridi. The regression model accounted for clustering of participants by village. A 5% significance threshold was adopted.

## Results

1. *Community-based surveys in Maridi County, 2018–2024*

In Maridi, community-based surveys to compare CDTI coverage during annual (2017) and bi-annual (2021 and 2023) schemes found a significant change in overall coverage between surveys, from 40.8% in 2017 to 70.3% in 2023 (Table 1). However, coverage data obtained through door-to-door interviews by the research team (surveyed coverage) revealed significantly lower ivermectin intake compared to the reported CDTI coverage (p<0.001). Comparing CDTI therapeutic coverage across villages, the 2024 survey revealed that there were no significant differences between the proportions of individuals in each site who received at least one dose of ivermectin in 2023: Hai-Gabat (70.6%) vs Kazana-1 (69.0%) vs Kazana-2 (71.2%); p=0.531.

Based on the survey data collected at specific study sites within Maridi County, it was observed that the proportion of the population that benefits from at least one dose of ivermectin annually has been increasing steadily over the years. Ivermectin intake increased in all population subgroups but most significantly in the 5–10-year age group (Table 2).

**Table 1. CDTI coverage in Maridi in 2017, 2021, and 2023.**

| Counties (year) | Persons surveyed | Persons who took ivermectin among those surveyed | Surveyed[a] therapeutic coverage | Reported[b] therapeutic coverage* |
|---|---|---|---|---|
| Maridi (2017) [11] | 17,652 | 7,209 | 40.8% | 82.3% |
| Maridi (2021) [25] | 14,378 | 8,134 | 56.6% | 93.6% |
| Maridi (2023) [2024 survey] | 4,395 | 3,088 | 70.3% | 74.3% |

*Data from the World Health Organization: https://espen.afro.who.int/diseases/onchocerciasis.

[a]Door-to-door verbal surveys, denominator was the entire surveyed population.

[b]Directly observed ivermectin intake by individuals, denominator was the total the population as per the pre-CDTI census (aggregated data for the entire Maridi County).

**Table 2. Ivermectin intake in 2017, 2021, and 2023 in Maridi study sites based on repeated community-based surveys (2017 & 2021 data extracted from [25]).**

| Year | 2017, one round | | 2021, two rounds | | 2023, two rounds | | p-value* |
|---|---|---|---|---|---|---|---|
| | Treated/Total | Coverage | Treated/Total | | Coverage | Treated/Total | Coverage |
| Total population | 7,209/17,652 | 40.8% | 8,134/14,378 | 56.6% | 3,088/4,395 | 70.3% | <0.001 |
| Females only | 3,794/9,206 | 41.2% | 4,180/7,512 | 55.6% | 1,574/2,279 | 69.1% | <0.001 |
| Males only | 3,406/8,445 | 40.3% | 3,928/6,833 | 57.5% | 1,512/2,110 | 71.7% | <0.001 |
| 5−10-year-olds | 904/3,485 | 25.9% | 1,683/3243 | 51.9% | 548/801 | 68.4% | <0·001 |
| 11−20-year-olds | 2,450/4,872 | 50.3% | 2,640/3,866 | 68.3% | 1,021/1,218 | 83.8% | <0·001 |

*Chi-squared test to investigate differences in proportions of ivermectin uptake across the three years being investigated.

Among participants who did not receive ivermectin in 2023, the reasons provided for not taking the drug were similar for both the May and November 2023 CDTI rounds. However, the proportion of those reporting that CDDs did not reach their home during the November 2023 round was significantly higher than in May 2023 (Table 3).

2. *Independent Coverage Evaluation Survey (CES) of ivermectin coverage in Maridi in 2021*

A total of 1,093 participants were surveyed in four payams (geographical zones) of Maridi County: Kozi (n = 93), Mambe (n = 112), Maridi (n = 737), and Ngamunde (n = 151). Overall, 509 participants reported taking ivermectin in August 2021 giving a CDTI therapeutic coverage of 46.6% (95% CI: 43.6-49.6). Among the CDTI non-compliant group (n = 326), the main reasons for not taking ivermectin included: being too far away during the CDTI (23.7%), being pregnant at the time of the CDTI (14.6%), being under-age (i.e., <5 years old; 10.6%), and CDDs not reaching the participants' homes (10.3%). The exhaustive list of the reasons for not taking ivermectin during the 2021 CES is provided in S2 File.

**Table 3. Reasons for not taking ivermectin during the two CDTI rounds of 2023.**

| Reasons for not taking ivermectin | May 2023 CDTI round (n = 1376) | November 2023 CDTI round (n = 1545) | P-value |
|---|---|---|---|
| Absent during CDTI | 345 (25.1%) | 417 (27.0%) | 0.243 |
| Less than 5 years old | 663 (48.2%) | 661 (42.8%) | 0.003 |
| Pregnant during CDTI | 91 (6.6%) | 79 (5.1%) | 0.084 |
| Breastfeeding during CDTI | 13 (1.0%) | 30 (1.9%) | 0.044 |
| Very sick during CDTI | 35 (2.6%) | 24 (1.6%) | 0.058 |
| Distributor did not reach my home | 164 (11.9%) | 263 (17.0%) | <0.001 |
| Refusal/ fear of side effects | 65 (4.7%) | 71 (4.6%) | 0.797 |

During the ivermectin CES of 2021, CDTI coverage was higher among males (256/496; 51.6%) than females (253/597; 42.4%), p = 0.003. Overall, 758 (69.4%) of participants acknowledged being aware of epilepsy and/or nodding syndrome (NS) in the community. Epilepsy and NS data were available for 817 participants. Of these, 49 (5.7%) and 19 (2.2%) self-reported having epilepsy and NS respectively, with 11 (1.3%) having both conditions. Considering the 57 persons with epilepsy and/or NS, their CDTI coverage was not significantly different from that of other participants (68.4% versus 61.2%, p = 0.346).

Participants' source of information about CDTI was also investigated among the 1,093 surveyed participants. The leading communication channel was through CDDs, community health workers and teachers (56.0%), followed by family members, friends and neighbours (32.4%), healthcare workers (13.0%), community and religious leaders (9.7%), radio (9.6%), posters (7.3%) and banners (0.6%). No participant remembered being informed about CDTI via social media, television, or flyers. Overall, 689 (63.0%) participants had received at least one source of information about CDTI prior to its implementation. The satisfaction rate regarding the CDTI programme was high, with 427/437 (97.7%) respondents being happy with the ivermectin treatment activity.

The multivariable logistic regression model revealed that all investigated covariates were significantly associated with ivermectin intake during the previous CDTI (Table 4). Children aged 5–14 years were more likely to take ivermectin compared to adults (15 years and above). Additionally, being male, being aware of the association between epilepsy and onchocerciasis, having access to multiple sources of CDTI information, and having epilepsy were all factors associated with higher likelihood of ivermectin intake.

Only participants aged ≥5 years and those with complete data were included in the regression model (n = 780). NS = nodding syndrome

3. *Ivermectin coverage survey in 2023 among children and their parents or guardians in Maridi*

The 2023 survey was targeting children and their adult caretakers in Maridi County, evaluating their ivermectin intake in the August 2022 CDTI round. Data from 386 individuals (253 children aged 3 − 9 years and 133 parents or guardians) were collected. Overall ivermectin therapeutic coverage in this population (August 2022) in Maridi was 43.8%, and it was higher among females (109/223; 48.9%) than males (60/163; 36.8%); p = 0.024. Among those aged ≥5 years, 185 (61.3%) had ever taken ivermectin. Previous ivermectin intake was reported in 7 (8.3%) of the 84 under-fives (3- and 4-year-olds), with no noticeable adverse events.

## CDTI perceptions and coverage in Maridi

The 2023 survey among children and their parents or guardians also explored perceptions about the CDTI in Maridi. Beliefs and perceptions vis-à-vis CDTI are summarised in Table 5. Based on the information provided by some of the

**Table 4. Multiple logistic regression investigating the determinants of ivermectin uptake in August 2021 in Maridi.**

| Covariates | Odds ratio (95% CI) | P-value |
|---|---|---|
| Age group | | |
| 5-14 years | Reference | |
| 15 years and above | 0.619 (0.529 – 0.723) | <0.001 |
| Gender | | |
| Female | Reference | |
| Male | 1.944 (1.569 – 2.409 | <0.001 |
| Awareness of epilepsy/ NS | 1.616 (1.029 – 2.537) | 0.037 |
| Number of sources of CDTI information | 1.381 (1.134 – 1.682) | 0.001 |
| Having epilepsy and/or NS | 1.612 (1.242 – 2.093) | <0.001 |

**Table 5. CDTI perceptions of individuals in Maridi, 2023.**

| Perceptions about ivermectin | n (%) | N |
|---|---|---|
| Take ivermectin systematically during every CDTI | 85 (74.6%) | 114 |
| Believe that ivermectin intake is beneficial for health | 127 (98.4%) | 129 |
| Believe that ivermectin intake can be problematic | 22 (17.5%) | 126 |
| Willing to take ivermectin during the next CDTI | 128 (98.5%) | 130 |
| Willing for my child to take ivermectin during the next CDTI | 129 (99.2%) | 130 |
| Willing to pick up ivermectin myself at distribution point | 127 (98.4%) | 129 |
| Know that ivermectin protects children from getting epilepsy | 112 (91.8%) | 122 |
| Participate in CDTI despite known pregnancy | 3 (60.0%) | 5 |

All study data are available at S3 File

adult respondents, about three quarters of parents (74.6%) reported taking ivermectin during every CDTI round when it is provided. Among these adult respondents, there were 89 women of child-bearing age (18−49 years), of whom 58 (65.2%) had taken ivermectin during the last CDTI in August 2022. Of the five women who knew that they were pregnant at the time of CDTI, three (60%) still took ivermectin and no unfavourable outcome was noted for them or their babies.

## Discussion

This study demonstrates a recent significant increase in treatment coverage following reinforced awareness campaign and introduction of biannual CDTI by the Government of the Republic of South Sudan. After switching to bi-annual CDTI in Maridi, overall coverage initially increased from 40.8% in 2017 to 56.6% in 2021, and then to 70.3% in 2023 based on the results of the community surveys. Achieving this increase in CDTI coverage in Maridi, albeit modest, still stands as a remarkable feat considering that most settings experienced drastic drops in ivermectin intake in the post-COVID era [27,28]. The increase in coverage in Maridi was also reflected in OAE-vulnerable groups, i.e., children aged 5–10 years. The fact that lower ages were associated with increased odds of taking ivermectin in the multivariable model is in stark contrast with previous observations [29] and could be the consequence of increased awareness about onchocerciasis and epilepsy in Maridi.

The fact that CDTI coverage increased only moderately in 2021 could be attributable to a lingering COVID-19 scare, in the aftermath of the pandemic [27,30]; however, CDTI adherence eventually improved with the persistent campaign leading to a much higher coverage in 2023. The ivermectin coverage for the year 2021 obtained during door-to-door investigations by Amref was 56.6% (55.8–57.4) in selected villages within the Maridi Central area, while that same year, a CES conducted in a larger area of Maridi found a 46.6% (95% CI: 43.6–49.6) CDTI coverage. This difference may be a consequence of Amref's research activities on the link between onchocerciasis and epilepsy in the Maridi Central area, which made residents of this site more likely to take ivermectin.

The difference between surveyed ivermectin coverage and the reported coverage by the government was previously observed in several onchocerciasis-endemic countries across Africa [29] and can be partially explained by a reporting bias by the CDDs or County NTD officers whose hierarchy would certainly be pleased with higher CDTI coverage numbers. Unregistered demographic changes (due to deaths, population movements, etc.) can also alter the denominator used for coverage calculations. In Cameroon, differences between reported and surveyed coverages were attributed, amongst other reasons, to an incomplete census leading to an underestimation of the total population or the (in)voluntary misre-porting of persons treated by the CDDs [31]. Therefore, the government reported coverage does not always reflect the reality and must be interpreted with caution. The 70% coverage achieved in 2023 is still below the 80% target to achieve interruption of transmission within reasonable timelines. Given the increasing ivermectin coverage because of continuous

sensitisation, it is likely that adequate CDTI coverage will be achieved and sustained in Maridi which will ensure steady progress toward onchocerciasis elimination.

The CES conducted in 2021 found a trend of higher ivermectin intake among persons with epilepsy compared to the general population (68.4% versus 61.2%), although the difference was not statistically significant. There is evidence supporting that ivermectin also can reduce the frequency of seizures in persons with epilepsy [32,33] which might be the motivating factor for its increased intake by seizure-prone individuals. It is unlikely that the "anti-seizure" effect of ivermectin results from the penetration of the drug into the central nervous system as it cannot cross the blood-brain barrier under normal conditions [34]. A more plausible explanation is that the reduction in the *O. volvulus* microfilarial load after ivermectin intake is associated with fewer seizures [8,33,35].

In 2023, another was CES conducted by CBM to confirm the reported CDTI coverage for the May-June 2023 distribution round in several counties. This 2023 CES found a survey CDTI coverage of 81% in Maridi, which was most likely an overestimation. Indeed, 22.7% of the participants in this CES were excluded from the denominator because they were absent during CDTI. Including these individuals would yield a more realistic CDTI coverage of 58.3% for May 2023 [36].

The Maridi 2023 survey of children and their parents or guardians revealed that some under-five children and pregnant women consciously took ivermectin, without any major adverse events reported. In the absence of adequate and well-controlled studies in humans, there is currently no evidence to confirm the safety of ivermectin in pregnancy or early childhood. However, in 2020 in France, ivermectin was approved for the treatment of helminthiasis during pregnancy irrespective of gestational age [37]. There is thus a precedent for prescribing ivermectin during pregnancy, and this is supported by the fact that poor materno-foetal outcomes have not been reported after exposure to ivermectin during pregnancy [38]. Moreover, it has been hypothesised that onchocerciasis treatment during pregnancy could spare the foetus from developing "parasite tolerance" towards *O. volvulus* which could foster early and intense onchocercal infection after birth [39]. Routinely including under-fives during CDTI should be considered because *O. volvulus* infections already happen in this age group, positioning these children as potential parasitic reservoirs that could stall elimination efforts [39,40]. In this light, clinical trials investigating the safety of ivermectin in pregnancy and early childhood are urgently warranted.

In most onchocerciasis-endemic areas in sub-Saharan Africa, there is little awareness about OAE within the community, including persons with epilepsy and healthcare workers. Indeed, a survey in two rural onchocerciasis foci in Cameroon with an epilepsy prevalence >4% found that 92% of the 249 community participants had never heard about the association between onchocerciasis and epilepsy, and the majority of them believed that the high prevalence of epilepsy in their village was due to witchcraft [41]. Hence, educating the village residents about the onchocerciasis-epilepsy link could go a long way to reduce epilepsy-related stigma [42]. At the policy level, OAE awareness can also motivate stakeholders to strengthen onchocerciasis elimination measures to reap the benefits of epilepsy prevention [43]. In Maridi, the switch from annual to bi-annual ivermectin distribution with improved therapeutic coverage resulted in >80% decrease in the incidence of OAE including nodding syndrome [25], much higher than the 50% decrease predicted during the protocol phase [20]. Similar drops in OAE incidence were also observed in other onchocerciasis-endemic settings like in the Mvolo County of South Sudan after introducing annual CDTI [28], and in the Mahenge area of Tanzania after switching to bi-annual CDTI [44].

The knowledge that OAE is preventable by proper ivermectin intake is still poorly disseminated in the study communities and among CDDs; whereas this information could be used as a motivating factor to increase ivermectin uptake. The generally accepted health belief model posits that the perceived severity of a health condition determines the perceived threat posed by that condition, which in turn plays an essential role in motivating health behaviour change [45]. In view of the broad clinical spectrum of onchocerciasis which goes beyond the commonly known skin and ocular symptoms to include neurological manifestations [4], and considering that South Sudan is an important OAE focus [12,13,46], increasing the communities' knowledge base about onchocerciasis-related morbidity may indeed improve their health-seeking behaviour and motivate them to become significantly more compliant during CDTI campaigns. Our study showed that by

the year 2023, nearly all caretakers of children (91.8%) in Maridi were aware that ivermectin could protect their children against epilepsy and 98.4% said they were willing to collect ivermectin themselves at a distribution point once it was available. Therefore, emphasis should be laid on improving health system performance and implementing optimal distribution approaches that would provide ivermectin to all eligible persons.

When comparing the operational implementation of the two CDTI rounds of 2023 in Maridi, one notable observation is the increased frequency of CDDs not reaching the homes of surveyed individuals to distribute ivermectin, from 11.9% (May 2023) to 17.0% (November 2023). A high rate (16.1%) of non-visits to households by CDDs was also observed in Cameroon [31]. This could be related to the long distances CDDs have to cover to visit each home [47,48], and could also infer disgruntlement among CDDs who may feel over-burdened, under-motivated, and devoid of supervisory support. Besides the failure of CDDs to deliver ivermectin to some households, other reasons included the absence of household residents when the CDD did visit, and avoidance of the drug for fear of side effects. Indeed, a Cochrane review of 29 studies found that the fear of adverse events is frequently cited by communities when it comes to ivermectin intake [49]. Adverse events are even more feared in settings that are co-endemic with loiasis (as in some parts of South Sudan [50]), because fatal outcomes have occurred following ivermectin treatment in persons heavily infected with *Loa loa* [51]. Therefore, it is crucial to understand the setting where CDTI is being implemented and motivate the CDDs accordingly. Another important aspect is the provision of evidence-based information to the population to mitigate their fears of taking ivermectin.

With over two-thirds of the South Sudanese population practising pastoralism [52] and many others involved in subsistence farming in different locations depending on the season, many individuals are often missed in their homes during CDTI owing to their outdoor activities. A qualitative study has suggested that CDTI uptake among pastoralist communities in South Sudan could be improved by adjusting the timing of treatment campaigns to take into account pastoralists' seasonal migration patterns, by involving pastoralist leaders in CDTI planning and mobilisation activities, by ensuring that (more) CDDs are recruited from within the target communities, and by addressing negative perceptions about ivermectin [52]. Most of these suggested solutions can also be considered for non-pastoralist communities, in addition to another strategy which would entail an extra round of school-based ivermectin distribution among children aged 5 – 15 years who are most at risk for OAE, six months after annual CDTI [53]. Furthermore, considerations should be made towards storing ivermectin that is not distributed during CDTI at health centres where it can be collected by those who missed the house-to-house distribution or who become eligible shortly after CDTI such as pregnant women after giving birth. For instance, the distribution of leftover ivermectin at the child vaccination center of the Maridi County Hospital to postpartum women recovered 5.3% of missed CDTI targets in 2024 [53]. To further increase ivermectin coverage, healthcare workers at health facilities also could routinely ask whether patients and their household members received ivermectin during the last campaign and offer the drug to those who missed CDTI [54]. However, these strategies will only be successful and sustainable if people are convinced about the benefits of taking ivermectin. In the meantime, to complement the CDTI campaigns, alternative activities to reduce onchocerciasis transmission such as the "Slash and Clear" vector control method which has been embraced by the Maridi community should be encouraged and scaled-up in other parts of the country [55].

We acknowledge some limitations of our study such as the verbal self-reporting approach for estimating ivermectin intake with no means of verifying the veracity of the data, and the potential risk for recall bias. There were also slight differences in survey methodologies, notably the exhaustive door-to-door approach of the Amref research team versus the independent CES conducted in November 2021. Nevertheless, our findings still provide important insights into the reporting, evolution and challenges of onchocerciasis control using CDTI in Maridi where parasitological, serological, and entomological studies all attest to high ongoing onchocerciasis transmission [22,56]. We also shed more light on the reasons for low ivermectin intake and put forth some interventions that can effectively drive CDTI coverage upward in resource-constrained settings like South Sudan. Such data are crucial for stakeholders to make informed decisions

and plan for future actions. The steady rise in CDTI therapeutic coverage during recent years is encouraging and depicts the possibility of successfully implementing onchocerciasis elimination measures even in challenging settings like South Sudan [53].

In conclusion, the onchocerciasis situation in South Sudan is far from achieving interruption of transmission by 2030 as outlined in the WHO roadmap [57]. Stronger leadership, increased funding, supportive supervision, implementation research and strategic interventions must be deployed to sustainably accelerate elimination prospects and decrease the disease burden caused by onchocerciasis in South Sudan. Our study showed that onchocerciasis awareness campaigns and bi-annual distribution of ivermectin can considerably increase CDTI coverage. It also points to possible ways to further optimise CDTI and sustainably achieve satisfactory coverage in Maridi

## Supporting information

**S1 File. Reasons for not taking ivermectin as recorded during the Coverage Evaluation Survey of 2021.**
(PDF)

**S2 File. Study questionnaire for assessment of ivermectin coverage.**
(DOCX)

**S3 File. Study data.**
(XLSX)

## Acknowledgments

We are grateful to the communities and authorities of the Maridi County in South Sudan for their participation and support during the various surveys.

## Author contributions

**Conceptualization:** Joseph Nelson Siewe Fodjo, Stephen Raimon Jada, Robert Colebunders.

**Data curation:** Joseph Nelson Siewe Fodjo.

**Formal analysis:** Joseph Nelson Siewe Fodjo.

**Funding acquisition:** Joseph Nelson Siewe Fodjo, Luis-Jorge Amaral, Robert Colebunders.

**Investigation:** Moses Okwii, Stephen Raimon Jada, Amber Hadermann, Jacopo Rovarini, Makoy Y Logora, Johan Willems.

**Methodology:** Joseph Nelson Siewe Fodjo, Luis-Jorge Amaral, Robert Colebunders.

**Project administration:** Johan Willems.

**Resources:** Joseph Nelson Siewe Fodjo, Stephen Raimon Jada, Jane Y Carter, Robert Colebunders.

**Supervision:** Joseph Nelson Siewe Fodjo, Stephen Raimon Jada, Robert Colebunders.

**Validation:** Joseph Nelson Siewe Fodjo.

**Visualization:** Joseph Nelson Siewe Fodjo.

**Writing – original draft:** Joseph Nelson Siewe Fodjo, Robert Colebunders.

**Writing – review & editing:** Joseph Nelson Siewe Fodjo, Moses Okwii, Stephen Raimon Jada, Amber Hadermann, Jacopo Rovarini, Luis-Jorge Amaral, Rogers Nditanchou, Yak Yak Bol, Makoy Y Logora, Jane Y Carter, Johan Willems, Robert Colebunders.

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
