## [Decision Letter · Decision Letter 0]

9 Jul 2025

Response to Reviewers
Revised Manuscript with Track Changes
Manuscript

Shaden Kamhawi

co-Editor-in-Chief

Paul Brindley

co-Editor-in-Chief

**Journal Requirements:**

At this stage, the following Authors/Authors require contributions: Robert Colebunders. Please ensure that the full contributions of each author are acknowledged in the "Add/Edit/Remove Authors" section of our submission form.

3) Some material included in your submission may be copyrighted. According to PLOSu2019s copyright policy, authors who use figures or other material (e.g., graphics, clipart, maps) from another author or copyright holder must demonstrate or obtain permission to publish this material under the Creative Commons Attribution 4.0 International (CC BY 4.0) License used by PLOS journals. Please closely review the details of PLOSu2019s copyright requirements here: PLOS Licenses and Copyright. If you need to request permissions from a copyright holder, you may use PLOS's Copyright Content Permission form.

Potential Copyright Issues:

- Please confirm (a) that you are the photographer of Supplementary File 1, or (b) provide written permission from the photographer to publish the photo(s) under our CC BY 4.0 license.

- Figures 1 and 3. Please (a) provide a direct link to the base layer of the map (i.e., the country or region border shape) and ensure this is also included in the figure legend; and (b) provide a link to the terms of use / license information for the base layer image or shapefile. We cannot publish proprietary or copyrighted maps (e.g. Google Maps, Mapquest) and the terms of use for your map base layer must be compatible with our CC BY 4.0 license.

4) In the online submission form, you indicated that "Data are available with the authors by a simple request". All PLOS journals now require all data underlying the findings described in their manuscript to be freely available to other researchers, either

- In a public repository

- Within the manuscript itself

- Uploaded as supplementary information.

5) Please ensure that the funders and grant numbers match between the Financial Disclosure field and the Funding Information tab in your submission form. Note that the funders must be provided in the same order in both places as well. State the initials, alongside each funding source, of each author to receive each grant. For example: "This work was supported by the National Institutes of Health (####### to AM; ###### to CJ) and the National Science Foundation (###### to AM).".

**Reviewers' comments:**

**Key Review Criteria Required for Acceptance?**

**Methods:**

-Are the objectives of the study clearly articulated with a clear testable hypothesis stated?

-Is the study design appropriate to address the stated objectives?

-Is the population clearly described and appropriate for the hypothesis being tested?

-Is the sample size sufficient to ensure adequate power to address the hypothesis being tested?

-Were correct statistical analysis used to support conclusions?

-Are there concerns about ethical or regulatory requirements being met?

Reviewer #1: Overall the methods are clearly stated and appropriate.

It is not mentioned if the survey approaches/field work needed to be adapted due to insecurity. If they did then this should be documented as an important contribution to these types of areas - which should also flow to results and methods.

Reviewer #2: see summary and general comments

**Results**

-Does the analysis presented match the analysis plan?

-Are the results clearly and completely presented?

-Are the figures (Tables, Images) of sufficient quality for clarity?

Reviewer #1: The results follow the methods well & clearly presented.

Figure 3 seems to be obtained from another source - can this be clarified.

Reviewer #2: see summary and general comments

**Conclusions**

-Are the conclusions supported by the data presented?

-Are the limitations of analysis clearly described?

-Do the authors discuss how these data can be helpful to advance our understanding of the topic under study?

-Is public health relevance addressed?

Reviewer #1: Overall the conclusions reflect the findings and present a range of important issues re social mobilisation, reporting daat/coverage surveys/ epilepsy. What is missing is how they did this or needed to adapt the strategies in a conflict/insecure environment. There is little information on this to fully appreciate the context.

Reviewer #2: see summary and general comments

**Editorial and Data Presentation Modifications?**

Reviewer #1: Minor

Reviewer #2: (No Response)

**Summary and General Comments**

Reviewer #1: Overall a well written and important paper on factor influencing the increase in MDA for onchocerciasis in South Sudan. It covers many important topics. More on the conflict/insecure environment and how it may have impacted any of the activities would be useful to include as this is a neglected area of NTD research and the more information that is gathered and published, the more that practical strategies may be developed from lessons learned. I don't envisage this addition to be extensive but needs more present to appreciate the context.

Reviewer #2: Fodjo et al. evaluated the impact of onchocerciasis awareness programs combined with biannual ivermectin treatment (CDTI) in the Maridi region of South Sudan. The study demonstrated that integrating reinforced community awareness with biannual CDTI significantly increased ivermectin treatment coverage. This combined approach proved more effective than standard strategies and presents a promising avenue for advancing onchocerciasis elimination efforts in remote and conflict-affected areas of South Sudan.

Major Concerns:

1) Several studies, including some from the same research group, have already reported on the outcomes of awareness programs and biannual CDTI in other countries. However, these publications are not cited, and no comparison is made between findings across different settings. A broader contextualization would strengthen the manuscript and highlight the relevance of the results beyond South Sudan.

2) The same research group previously published a study protocol for biannual CDTI in the same region, but this earlier work is neither cited nor discussed. The manuscript should clarify the relationship between the current study and the published protocol.

3) Figure 3 is blurry and difficult to interpret. A higher-resolution version should be provided to ensure clarity and readability.

4) The full questionnaires used in the study should be included in the supplementary materials to allow for transparency and reproducibility.

5) The authors note that compliance with ivermectin treatment is higher among individuals affected by nodding syndrome (NS) or epilepsy. It would be important to also address other clinical manifestations of onchocerciasis, such as blindness or hyperreactive onchodermatitis. How many individuals in the study area suffer from these or other onchocerciasis-related symptoms? Is treatment compliance similarly higher among these individuals?

PLOS authors have the option to publish the peer review history of their article (what does this mean? ). If published, this will include your full peer review and any attached files.

**Do you want your identity to be public for this peer review?** For information about this choice, including consent withdrawal, please see our Privacy Policy .

Reviewer #1: No

Reviewer #2: No

**Figure resubmission:****Reproducibility:** To enhance the reproducibility of your results, we recommend that authors of applicable studies deposit laboratory protocols in protocols.io, where a protocol can be assigned its own identifier (DOI) such that it can be cited independently in the future. Additionally, PLOS ONE offers an option to publish peer-reviewed clinical study protocols. Read more information on sharing protocols at https://plos.org/protocols?utm_medium=editorial-email&utm_source=authorletters&utm_campaign=protocols

---

## [Editor Report · Decision Letter 1]

20 Aug 2025

Dear DR COLEBUNDERS,

We are pleased to inform you that your manuscript 'Community-directed treatment with ivermectin in Maridi, South Sudan: Impact of an onchocerciasis awareness campaign and bi-annual treatment on therapeutic coverage' has been provisionally accepted for publication in PLOS Neglected Tropical Diseases.

Best regards,

Angela Monica Ionica, Ph.D.

Academic Editor

Qu Cheng

Section Editor

Shaden Kamhawi

co-Editor-in-Chief

Paul Brindley

co-Editor-in-Chief

---

## [Editor Report · Acceptance letter]

Dear Prof Colebunders,

We are delighted to inform you that your manuscript, " 

Community-directed treatment with ivermectin in Maridi, South Sudan: Impact of an onchocerciasis awareness campaign and bi-annual treatment on therapeutic coverage ," has been formally accepted for publication in PLOS Neglected Tropical Diseases.

Best regards,

Shaden Kamhawi

co-Editor-in-Chief

Paul Brindley

co-Editor-in-Chief
